# The Selectively Nontargeted Analysis of Halogenated Disinfection Byproducts in Tap Water by Micro-LC QTOFMS

**DOI:** 10.3390/toxics12090630

**Published:** 2024-08-26

**Authors:** Jing Wu, Yulin Zhang, Qiwei Zhang, Fang Tan, Qiongyu Liu, Xiaoqiu Yang

**Affiliations:** 1School of Optoelectronic Materials & Technology, Jianghan University, Wuhan 430056, China; 19503826580@163.com (J.W.); 18238778284@163.com (Y.Z.); keewezhang@jhun.edu.cn (Q.Z.); yanxizaof@sina.com (F.T.); 2College of Environment and Health, Jianghan University, Wuhan 430056, China

**Keywords:** selectively nontargeted analysis, halogenated disinfection byproducts, tap water, QTOFMS

## Abstract

With the rapid development of society, more and more unknown halogenated disinfection byproducts (DBPs) enter into drinking water and pose potential risks to humans. To explore the unknown halogenated DBPs in tap water, a selectively nontargeted analysis (SNTA) method was developed by conducting micro-liquid chromatography coupled with quadrupole time-of-flight mass spectrometry (micro-LC-QTOFMS). In this method, two runs were employed: in the first run, the modes of *TOFMS* and *precursor ion* (the fragments were set as Cl^35^/Cl^37^, Br^79^/Br^81^, and I^126.9^) were performed, and the molecular ions or precursor ions of the halogenated organics could be obtained; in the second run, the *product ion* mode was conducted by setting the molecular ion screened above, and the MS/MS spectrums could be acquired to speculate concerning the structure. Two kinds of model DBPs (one kind had an aliphatic structure and the other was an aromatic compound) were used to optimize the parameters of the MS, and their MS characteristics were summarized. With this SNTA method, 15 halogenated DBPs were screened in two tap water samples and their structures were proposed. Of them, six DBPs had not been reported before and were assumed to be new DBPs. Overall, the detected halogenated DBPs were mostly acidic substances.

## 1. Introduction

To acquire safe drinking water, chemical disinfection (e.g., HOCl, NH_2_Cl, and ClO_2_) is essential to inactivate pathogen microorganisms. A chemical disinfectant could react with organic matter in the source water and form unintended disinfection byproducts (DBPs). Due to the widespread use of chlorine-containing disinfectant, chlorinated DBPs are very common. Moreover, bromine/iodine-containing compounds in raw water or bromine/iodine impurities in chlorine-containing disinfectants would bring a wider variety of halogenated DBPs [1,2,3,4]. It is reported that about 88% are halogen-containing compounds, among the more than 6000 reported DBPs [5]. These halogenated DBPs in drinking water may bring health risks of cytotoxicity [6], mutagenicity [7], and carcinogenicity [8] to humans. With the rapid development of agriculture and industry, more and more environment pollutants enter into source water [9], which may be transferred into unknown halogenated DBPs during the disinfection procedure. Therefore, it is essential to develop a nontargeted analysis (NTA) method to explore these potentially threatening compounds.

Utilizing the powerful separation and identification functions of chromatography and mass spectrometry (MS) [10], NTA methods can detect as many organics as possible and play an important role in discovering unknown new DBPs [11,12]. A full scan MS was the most commonly used data acquisition mode in the NTA, which could theoretically detect all ions entering the mass analyzer [13,14]. However, due to the limit scanning frequency and data parsing ability, the compounds with a lower abundance could not be detected or the signals might be masked by higher abundance compounds. To compensate for the above shortcomings, some selective data analysis or data acquisition methods in the NTA were developed, and these methods could be called a selective nontargeted analysis (SNTA). In a work on the nontargeted screening of chlorinated DBPs formed from natural organic matters, a halogen extraction code was developed to check the isotopic patterns of chlorinated DBPs and acquire accurate *m*/*z* formulas of chlorinated DBPs [12]. To achieve the goal of a fast, accurate screening of halogenated DBPs, Zhang’s group developed a fast and selective method and conducted a series of magnificent works in exploring halogenated DBPs [15,16,17,18,19]. This method was conducted by a precursor ion scan (PIS) using triple quadrupole mass spectrometry (TQMS) [15,16], and the workflow of PIS was as follows: first, the full scan was conducted in the appropriate MS parameters to acquire the molecular ions of all the components in the sample; second, the precursor ion scan was processed by setting the mass of Cl, Br, or I to screen the components containing halogen elements; finally, the *product ion* scan was employed to acquire the MS/MS information for speculating concerning the structure of the halogenated components screened. From 2008 to 2019, about 150 halogenated DBPs were screened and identified using this PIS method [20].

Up until now, the PIS method was mostly conducted by TQMS with or without ultra-performance liquid chromatography (UPLC). TQMS was a kind of low-resolution MS. For some complex or new compounds, the identification needed an auxiliary of a high-resolution MS, such as time-of-flight mass spectrometry (TOFMS) [20]. Quadrupole TOFMS (QTOFMS) has similar functions to TQMS and has a higher resolution comparatively. As a result, it is assumed that the PIS method could be conducted by QTOFMS and may be better in screening and identifying halogenated DBPs. In addition, few works have been carried out on the NTA analysis of DBPs in the tap water of Wuhan city. In view of the above status, a SNTA method for halogenated DBPs with Micro-LC-QTOF-MS was developed. Some halogenated chain and aromatic DBPs were chosen as the models to optimize the instrumental parameters and to validate the method. The SNTA method was applied to two tap waters of Wuhan city, and three kinds of solid phase extraction (SPE) cartridges were utilized to obtain as many halogenated DBPs as possible.

## 2. Materials and Methods

### 2.1. Chemicals, Reagents, and Materials

The EPA (Environmental Protection Agency) 552 haloacetic acid mixed solution (trichloroacetic acid, chloroacetic acid, bromoacetic acid, dichloroacetic acid, 2,4-dichlorophenol, dibromo acetic acid, and 2,4,5-trichlorophenol, with each being 1000 mg/L in tetramethyl tert-butyl ether), 2,3,4,5-tetrachloro-1,2-benzoquinone, and 2,3-dibromo-5,6-dimethyl-1,4,-benzoquinone were all purchased from Accu Standard (New Haven, CT, USA); the 2-chloro-1,4-benzoquinone was obtained from Macklin Biochemical Technology Co., Ltd. (Shanghai, China); the 2-monoiodo-1,4-benzuoquinone (≥98%), monoiodoacetic acid (98%), diiodoacetic acid (98%), and bromoiodoacetic aced (98%) were purchased from CFW Laboratories, Inc. (Newark, DE, USA); the acetonitrile of HPLC grade was obtained from Sinopharm Chemical Reagent Co., Ltd. (Shanghai, China); the methanol of LC-MS grade was purchased from Thermo Fisher Scientific (Waltham, MA, USA); and the model DBPs’ mixed solution was composed of the haloacetic acid mixed solution, four halobenquinones, and three iodinated acetic acids, with each being 500 mg/L, dissolved in acetonitrile. The SPE cartridges of Bond Elut ENV (200 mg/6 mL) and Bond Elut C18 (200 mg/6 mL) were purchased from Agilent Technologies, Inc. (Santa Clara, CA, USA), and the Oasis HLB (200 mg/6 mL) cartridges were from Waters (Milford, MA, USA).

### 2.2. Sample Collection and Pretreatment

Two tap water samples and the corresponding source water were collected. One was taken from the tap of our laboratory fed by a drinking water plant with the Yangtze River as its source water, and the other came from a consumer’s tap supplied by a water plant with the Han River as the source water. The source water was taken from the input water of the corresponding drinking water plant. All the samples were collected in 4 L amber glass bottles with Teflon-lined screw caps, were shipped with ice to the laboratory, and were stored in a refrigerator at 4 °C. The SPE pretreatment, including three kinds of cartridges (ENV, HLB, and C18), was slightly adjusted based on the literature [14], and the detailed process is demonstrated in Appendix A. To determine whether there were any impurities in the solvents or any error in the sample collection and pretreatment, a control sample was performed using the same procedure with 4 L ultrapure water from a Milli-Q Type 1 water system (Millipore, Burlington, MA, USA).

### 2.3. Instrumental Method

A micro-LC system (M5, SCIEX, Palo Alto, CA, USA) was used to conduct the chromatography separation with a HSS T3 column (1.8 mm particle size, 1.0 mm × 100 mm, Waters, USA). The parameters of the micro-LC were adjusted based on the literature [14]. In total, 5 μL of the pretreatment sample was injected in the micro-LC and the column temperature was 40 °C. The mobile phase A was a mixed solution of 2% acetonitrile and 98% water; the mobile phase B was 98% acetonitrile with 2% water; the flow rate was 15 μL/min; and the gradient eluent was 0–3 min, 10% B; 3–30 min, 10–90% B; 30–36 min, 90–10% B; and 40–55 min, 10% B. 

A QTOFMS (Triple TOF 5600+, SCIEX, Palo Alto, CA, USA) with an electrospray ionization (ESI) source in negative mode was employed to conduct the data acquisition. Two runs were performed on each sample: first, a method in *TOFMS* mode (*m*/*z* 30–600 Da) coupled with *precursor ion* mode (the fragments were set as Cl^35^/Cl^37^, Br^79^/Br^81^, and I^126.9^) was conducted to collect the molecular information and screen the halogenated components; according to the retention time of the precursor ion spectrum, the corresponding molecular ions could be obtained from the TOFMS spectrum; then, the molecular ions screened in the first run were inputted into the list in *product ion* mode, and the MS/MS spectra were collected in the second run. The default value of most MS parameters was used, except for the delustering potential (DP) and collision energy (CE), which were optimized with two kinds of model halogenated DBPs. The parameters of the instrument were set as follows: source temperature: 250 °C; ion spray voltage: −4500 V; gas pressure of GS1: 16 psi, GS2: 17 psi, and curtain gas: 30 psi; DP: 20 V in all of the data acquisition mode; and CE: 6 V in *TOFMS* mode and 20 V in *precursor ion* and *product ion* modes.

## 3. Results and Discussion

### 3.1. Optimization of MS Parameters with Model DBPs

The EPA 552 haloacetic acid mixed solution, four halobenquinones, and three iodinated acetic acids were chosen as the model compounds to optimize the MS parameters. Natural organic matter is a complex mixture of diverse groups of organic compounds, humic and fulvic acids, proteins, peptides, and so on [21,22,23]. These substances are rich in aromatic rings and are easily decomposed to aromatic compounds after water treatment [24]. In the 6000 reported DBPs, aliphatic-based and aromatic were the two main carbon skeletons [5]. In addition, haloacetic acids and benzoquinones were the two common polar DBPs in drinking water [20,25,26]. For the MS parameters, the DP and CE were two important factors for the intensity of the molecular ions and the corresponding fragments, so they were optimized in the series of experiments by the model DBPs. 

Figure 1 demonstrates the effects on the molecular ions’ intensity of model DBPs in various DP values. The DP mainly acts on the needle pip of the ESI source to form an electrostatic field preventing the aggregation cluster of ions from the target compounds. An appropriate DP can improve the analytical sensitivity, while an excessive DP possibly induces analyte decomposition in the source. From Figure 1, it can be seen that the change of the DP value from 10 to 90 brings similar effects on the two kinds of model DBPs (Figure 1A: haloacetic acids; and Figure 1B: halophenols and halobenzoquinones). Most of the molecular ion’s intensity achieved was the highest at 20–30 V. Taking into account the lifetime of the instrument and the experimental results, 20 V was chosen as the optimized DP value.

Figure 2 exhibits the effects of various CE values on the fragments of model DBPs in *product ion* mode under the DP value of 20 V. It can be seen that, for haloacetic acids (HAAs, Figure 2A), the optimum CE that corresponds to the maximum Cl^−^, Br^−^, and I^−^ is in the range of 15–35 V; for halophenols and halobenzoquinones (Figure 2B), the optimum CE is in the range of 20–30 V. Overall, 20 V is appropriate because most of the fragmented halogen ions that form the model’s HOCs have a strong S/N. Moreover, the intensity of most molecular ions is still strong under the CE of 20 V (Appendix A). This optimized parameter was also suitable for the *precursor ion* mode, because the appropriate abundance of the fragments and molecular ions were beneficial for screening the halogen-containing substances. Appendix A presents the mass spectra of the model DBPs acquired in *precursor ion* mode, where, it can be seen, all of the molecular ions were found. BIAA could be screened not only in the precursor spectra of Br^79^ (Appendix A) and Br^81^ (Appendix A) but also in that of I^126.9^ (Appendix A), because this compound contains Br and I simultaneously.

With the MS/MS spectra obtained in the *TOFMS* and *product ion* mode under the optimized MS parameters, the MS and chromatograph characteristics of the model DBPs were obtained and listed in Table 1 and Table 2. This information was expected to provide a reference for the identification of the unknown DBPs. For the two kinds of DBPs, HAAs have the shortest retention time (RT), followed by the aromatic compounds of halobenzoquinones and halophenols. The abundance ratios of the isotopic ions in the MS spectra were consistent with the theoretical calculation discussed in the literature [20] and could help with the interpretation of the number of halogen atoms in unknown DBPs. For the fragments acquired in *product ion* mode, it could be seen that (M-H-CO_2_)^−^ was the common fragment of HAAs, and (M-H-HX)^−^ was the frequent fragment of halophenols. The fragments of halobenzoquinones were relatively complicated: (M-HX)^−^, (M-HX-CO)^−^, (M-HX-CO_2_)^−^, and (M-HX-CO_2_-CO)^−^ were the possible fragments occurring in the MS/MS spectra.

### 3.2. Detection of Halogenated DBPs in Tap Water with SNTA Method

Two tap water samples and the corresponding source water were grabbed to detect halogenated DBPs with the established SNTA method, and the two samples were named as T1 and T2, respectively. The source water of T1 was from the Yangtze River, and the source water of T2 was from the Han River. The two water samples were all pretreated with three SPE cartridges (HLB, ENV, and C18). Comparing with the halogenated organic compounds detected in the source water, the ones that were unique to tap water were assumed to be halogenated DBPs and are listed in Table 3. Four DBPs were confirmed through the MS spectra and retention time with the model DBPs (MCA, DCAA, and TCAA; their chromatographs and MS spectra are demonstrated in Appendix A) or the purchased standard (bromochloroacetic acid, Appendix A). MCA, DCAA, and TCAA are on the regulated list of DBPs in various countries and regions, and bromochloroacetic acid is one of the common haloacetic acid DBPs [25]. The other DBPs were proposed by monoisotopic mass and isotope characteristics obtained by TOF-MS spectra combined with the fragment ions from the MS/MS spectra conducted in *product ion* mode. 

The deduction procedure was exemplified by the compound of 2-(1-amino-1-chloro-2-(4-nitropheyl) ethyl) malonic acid with ion cluster *m*/*z* 300.8473/302.8407. According to the TOFMS spectra of 300.8473/302.8407 (Appendix A), it could be inferred that the chemical formula was C_11_H_11_ClN_2_O_6_. The double bond equivalent of this compound was calculated as 7, indicating that it might be an aromatic compound. The two losses of 44 (300.8473→256.8571 and 256.8571/258.8535→212.8627) indicated that the compound might contain two carboxyl groups. The difference of 30 between 256.8571 and 226.8405 meant that it might be a nitrobenzene compound. Additionally, the short retention time of 3.7 min meant that it was a highly polar DBP and might contain an amino group. The proposed structure of the ion cluster 300.8473/302.8407 and its fragmentation scheme are demonstrated in Figure 3. The most stable counterpoint structure of the two substituent groups on the benzene ring was chosen, although it might also be an adjacent or intermediate structure. Using the self-constructed web database of DBPs [5] and the latest literatures about DBPs, it was found that this DBP has not been reported before. Therefore, it was classified as a new DBP and was marked with # in the compound list in Table 3. 

The other eight ion clusters (*m*/*z* 193.8864/195.9109/197.8108/199.8036, *m*/*z* 190.9587/192.9563, *m*/*z* 229.1074/231.0884, *m*/*z* 267.1217/269.1199/271.1162, *m*/*z* 224.9189/226.9145, *m*/*z* 171.0206/173.0408, *m*/*z* 138.0385/140.0352, and *m*/*z* 286.0484/288.0526) with available MS/MS spectra were elucidated, and the procedures are depicted in Appendix A with the corresponding MS spectra and proposed structure being demonstrated in Appendix A. The chemical formulas of two ion clusters (164.9353/166.9307/168.9307 and 198.8751) without available MS/MS spectra were proposed through the molecular weight with a mass error smaller than 0.05 Da. Inputting the two formulas in the DBP database [5], it was found that the two formulas were in the list of known DBPs, and the corresponding DBP name with the literature are listed in Table 3.

In the fifteen halogenated DBPs detected in this work, six compounds were assumed to be new DBPs by comparing with the DBP database and the latest literature about DBPs. There was no significant difference in the detection frequency of the 15 DBPs between the two water samples (T1 and T2), although the source water of these two tap water samples was different. The precursor for these DBPs’ formation might be similar because the disinfectant using in both drinking water treatment plants was sodium hypochlorite. In addition, for the three SPE cartridges used for the pretreatment of the water samples, the DBPs detected using C18 were slightly more than the other two cartridges. Except for *m*/*z* 267.1217/269.1199/271.1162 and *m*/*z* 171.0206/173.0408, all the other DBPs could be measured in the samples with a C18 cartridge. Therefore, a C18 cartridge was more universal in the SNTA of the halogenated DBPs. Of the 15 screened halogenated, 13 were acidic compounds. This was consistent with the summary result of the function group about reported DBPs: acids were the largest category and accounted for 34% of the 20 categorized functional groups [5]. In addition, four halogenated nitrogenous DBPs should be paid more attention. It is reported that nitrogenous DBPs were forcing agents in the cytotoxicity of disinfected water among known DBPs [34,35]. 

## 4. Conclusions

In this work, a SNTA method for the screening of halogenated DBPs was developed with a micro-LC QTOFMS. This method was conducted with multiple data acquisition modes in two runs: first, the *TOFMS* in *precursor ion* mode was employed to acquire the precursor ions of the halogenated compounds; then, the mode of *product ion* was operated to obtain the MS/MS spectra for speculating concerning the structures. Two kinds of common halogenated DBPs (eight haloacetic DBPs and six aromatic DBPs) were chosen to optimize the MS parameters (the DP was set at 20 V in all modes; and the CE was set at 6 V in the *TOFMS* mode and at 20 V in the *precursor ion* and *product ion* modes), and their MS characteristics were summarized to provide a reference for the identification of unknown DBPs. This method was applied to two tap waters, and 15 halogenated DBPs were screened and their structures were proposed. Most of the screened halogenated DBPs were acid compounds and six of them were assumed to be new DBPs.

## Figures and Tables

**Figure 1 toxics-12-00630-f001:**
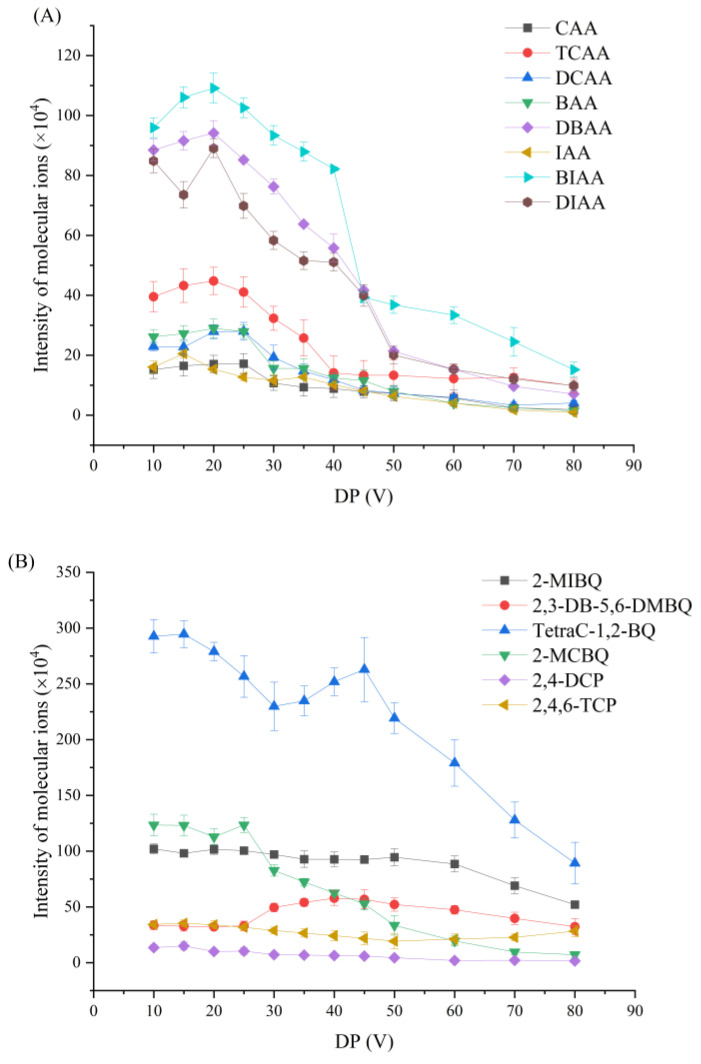
Optimization of delustering potential (DP) in *TOFMS* mode: (**A**) haloacetic acids—CAA: monochloroacetic acid; TCAA: trichloroacetic acid; DCAA: dichloroacetic acid; BAA: monobromoacetic acid; DBAA: dibromoacetic acid; IAA: monodiodoacetic acid; and BIAA: bromodiodoacetic acid; DIAA: diiodoacetic acid; and (**B**) halophenols and halobenzoquinones—2-MIBQ: 2-monoiodobenzoquinone; 2,3-DB-5,6-DMBQ: 2,3-dibromo-5,6-dimethylbenzoquinoen; Tetra C-1,2-BQ: tetrachloro-1,2-benzoquinone; 2-MCBQ: 2-monochlorobenzoquinone; 2,4-DCP: 2,4-dichlorophenol; and 2,4,6-TCP: 2,4,6-trichlorophenol.

**Figure 2 toxics-12-00630-f002:**
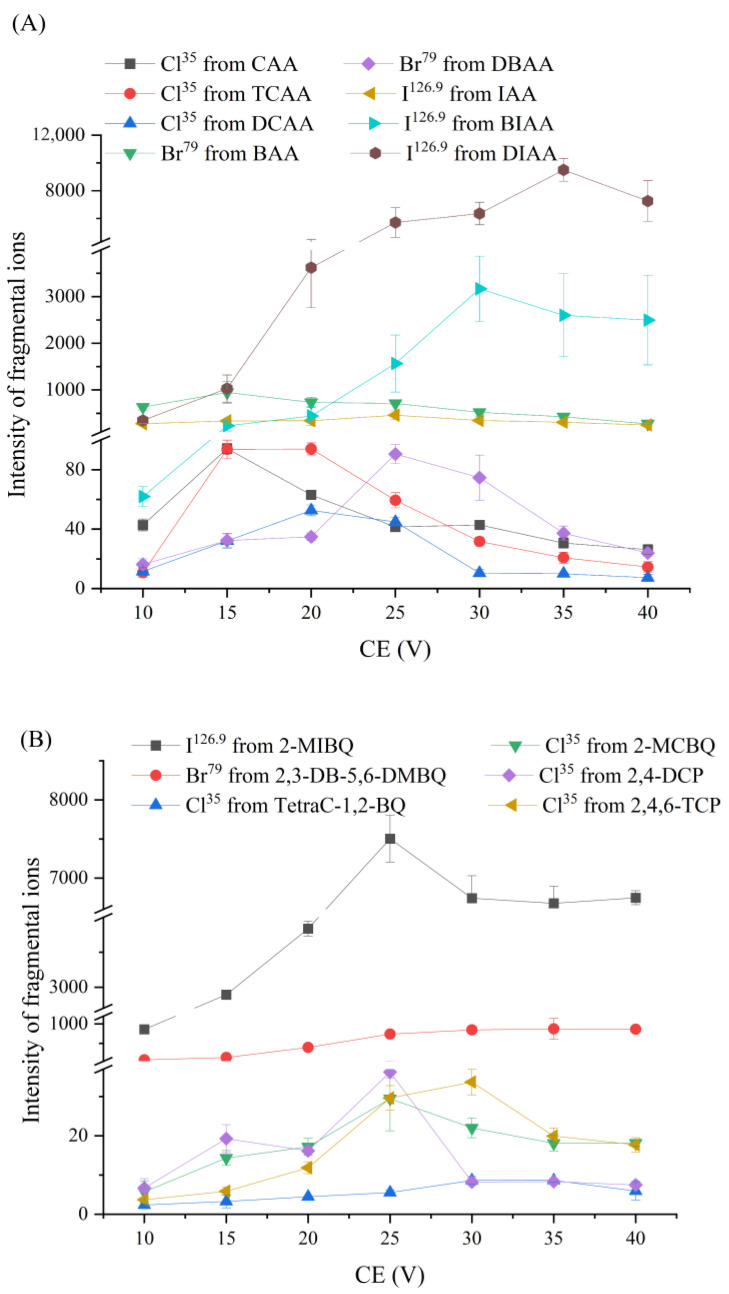
Optimization of collision energy (CE) in *product ion* mode: (**A**) haloacetic acids, and (**B**) halophenols and halobenzoquinones.

**Figure 3 toxics-12-00630-f003:**
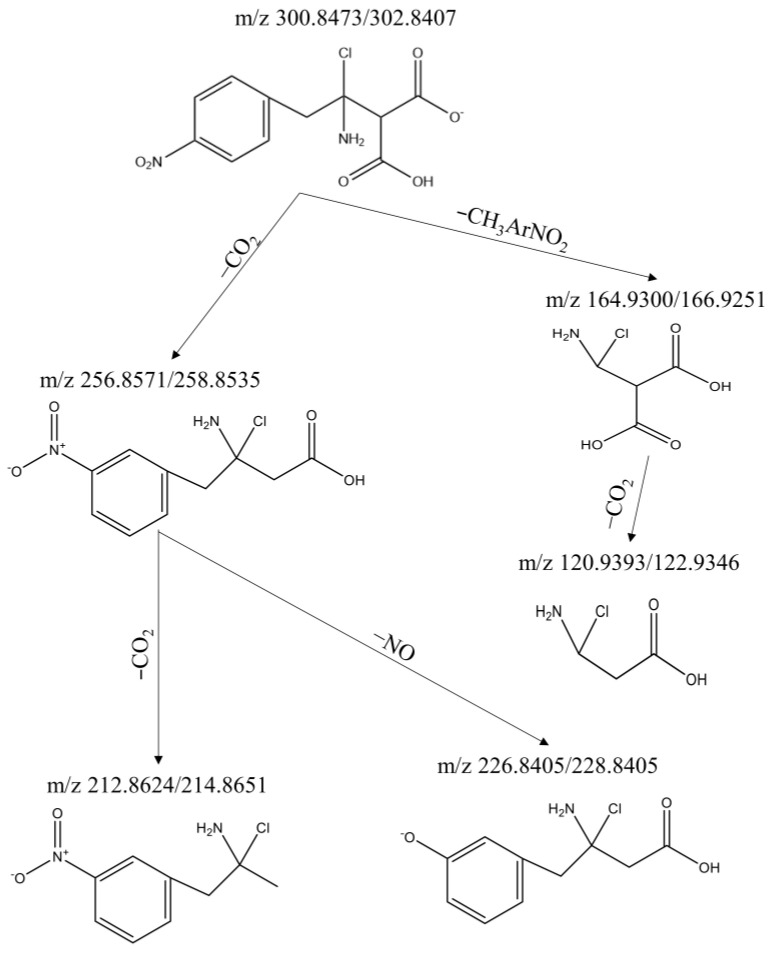
Proposed structure and fragmentation scheme for ion cluster 300.8473/302.8407.

**Table 1 toxics-12-00630-t001:** Characterization of MS spectra and chromatography of HAAs and halophenols.

	(M-H)^−^	(M-H+2)^−^	(M-H+4)^−^	(M-H+6)^−^	Ratio	(M-H-CO_2_)^−^	(M-H-HX)^−^	RT (min)
CAA	92.9757	94.9800			3:1	-		6.153
DCAA	126.9358	128.9324	130.9301		9:6:1	82.9464		7.489
TCAA	160.8911	162.8961	164.9016	166.9101	3:3:1:0.1	116.9078		2.239
BAA	136.9244	138.9224			1:1	93.0054		6.643
DBAA	214.8348	216.8401	218.8341		1:2:1	170.8452		9.556
IAA	184.9099					140.9189		8.754
BIAA	262.8214	264.8185			1:1	218.8214		11.947
DIAA	310.8072					266.8172		13.568
2,4-DCP	160.9574	162.9555	164.9516		9:6:1		124.9810	19.465
2,4,6-TCP	194.9186	196.9160	198.9129	200.9117	3:3:1:0.1		158.9358	20.882

Note that M: molecular; RT: retention time; CAA: monochloroacetic acid; TCAA: trichloroacetic acid; DCAA: dichloroacetic acid; BAA: monobromoacetic acid; DBAA: dibromoacetic acid; IAA: monodiodoacetic acid; BIAA: bromodiodoacetic acid; DIAA: diiodoacetic acid; 2,4-DCP: 2,4-dichlorophenol; and 2,4,6-TCP: 2,4,6-trichlorophenol.

**Table 2 toxics-12-00630-t002:** Characterization of MS spectra and chromatography of halobenzoquinones.

Compound	(M+H)·	(M+H+2)·	(M+H+4)·	(M+H+6)·	Ratio	X	(M-HX)^−^	(M-HX-CO_2_)^−^	(M-HX-CO_2_-CO)^−^	(M-HX-CO)^−^	RT (min)
2-MCBQ	142.9911	144.9888			3:1		107.0149	79.0202			14.584
2-MIBQ	234.9273					126.9038					15.687
2,3-DBBQ	312.8217	314.8225	316.8213		1:2:1		276.8511	232.9025	198.9249		19.404
Tetra C-1,2-BQ	244.8747	246.8755	248.9723	250.8964	8:10:5:1		208.8971			180.9016	18.467

Note that 2-MIBQ: 2-monoiodobenzoquinone; 2,3-DB-5,6-DMBQ: 2,3-dibromo-5,6-dimethylbenzoquinoen; Tetra C-1,2-BQ: tetrachloro-1,2-benzoquinone; and 2-MCBQ: 2-monochlorobenzoquinone.

**Table 3 toxics-12-00630-t003:** Halogenated DBPs screened from two tap waters.

Compound	Ion Cluster (*m*/*z*)	Formula	RT (min)	HLB-T1	HLB-T2	ENV-T1	ENV-T2	C18-T1	C18-T2
TCAA	160.8931/162.8961/164.9016/166.9101	C_2_HCl_3_O_2_	2.5	√	√	√	√	√	√
4,4-dichlorocyclobutene-1- carboxylic acid * [27]	164.9353/166.9307/168.9307	C_5_H_4_Cl_2_O_2_	3.7	×	×	√	√	√	√
2-(1-amino-1-chloro-2-(4-nitropheyl) ethyl) malonic acid *^,#^	300.8473/302.8407	C_11_H_11_ClN_2_O_6_	3.7	√	√	√	√	√	√
3-iodopropanoic acid * [16]	198.8751	C_3_H_5_IO_2_	4.1	×	×	√	√	√	√
N,2,2,2-Tetrachloroacetamide ^#^	193.8864/195.9109/197.8108/199.8036	C_2_HCl_4_NO	4.4	√	√	√	√	√	√
MCAA	92.9752/94.9757	C_2_H_3_ClO_2_	5.9	√	√	√	√	√	√
DCAA	126.9349/128.9342/130.9301	C_2_H_2_Cl_2_O_2_	7.3	√	√	√	√	√	√
bromochloroacetic acid	170.8919/172.8921/174.886	C_2_H_2_BrClO_2_	7.6	√	√	√	√	√	√
(Z)-3-bromo-4-oxopent-2-enoic acid * [28,29]	190.9587/192.9563	C_5_H_5_O_3_Br	14.7	×	×	×	×	√	×
2-(5-Chloro-2-hydroxy-3-(hydroxymethyl)-4-methylphenyl) acetic acid * [30]	229.1074/231.0884	C_10_H_11_ClO_4_	15.2	×	√	×	√	×	√
(2Z,5E)-4-bromo-2-chlorohepta-2,5-dienedioic acid *^,#^	267.1217/269.1199/271.1162	C_7_H_6_ClBrO_4_	17.1	√	×	×	×	×	×
3-[(2-Bromoethyl)sulfanyl]-2-oxopropanoic acid *^,#^	224.9189/226.9145	C_5_H_7_BrO_3_S	18.4	×	√	×	×	×	√
chlorosalicylic acid * [31,32]	171.0206/173.0408	C_7_H_5_ClO_3_	20.4	√	×	×	×	×	×
(1-amino-1-chloropropoxy)-methanol *^,#^ [33]	138.0385/140.0352	C_4_H_10_ClNO_2_	21.2	√	√	√	√	√	√
2-chloro-5-(3-(hydroxymethy)-5-nitrophenyl) pentanoic acid *^,#^	286.0484/288.0526	C_12_H_14_ClNO_5_	30.2	√	√	√	√	√	√

Note that *, or its isomers; ^#^, new DBPs; HLB, ENV, and C18 were three kinds of solid phase extraction cartridges; T1, tap water sample 1 ^#^; T2, tap water sample 2 ^#^; √, detected; and ×, not detected.

## Data Availability

The original data presented in the study are included in the article/Appendix A; further inquiries can be directed to the corresponding author.

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
