# Peer review of "The Selectively Nontargeted Analysis of Halogenated Disinfection Byproducts in Tap Water by Micro-LC QTOFMS"

_toxics, 2024, doi:10.3390/toxics12090630_

Round 1

Reviewer 1 Report

Comments and Suggestions for Authors

The aim of the research described in the manuscript was to develop a method for identifying disinfection by-products in tap water. The conclusions do not answer this question. They should be completed once again, providing a brief description of the method and the recommended operating parameters of the measuring device. A good article should not be longer than 25 pages. I suggest shortening chapters: 2,3,4,7,9.

Author Response

Comments 1: The aim of the research described in the manuscript was to develop a method for identifying disinfection by-products in tap water. The conclusions do not answer this question. They should be completed once again, providing a brief description of the method and the recommended operating parameters of the measuring device.

Response 1: Thank you for pointing this out. We agree with this comment. We have revised the conclusion in red font according to the suggestion in line 257-262 of page 9.

Comments 2: A good article should not be longer than 25 pages. I suggest shortening chapters: 2,3,4,7,9

Response 2: Thank you for suggestion. However, I'm confused with this comment because there is only 10 pages in our article including references, and there is no chapters of 2,3,4,5,9.

Reviewer 2 Report

Comments and Suggestions for Authors

The article entitled: "Selectively Non-Targeted Analysis of Halogenated Disinfection Byproducts in Tap Water by Micro-LC QTOFMS", submitted to Toxics is good, but some modifications are needed to be accepted by the journal. Here are some suggestions:

1) Keywords need to start with a capital letter.

2) The introduction has the most recent article in the year 2021. This cannot happen in the last 3 years (2022, 2023 and 2024) many researches were carried out with such proposed themes, please review this in detail, dear authors.

3) In item 2.2, I suggest an illustrative scheme of how the collection was done. The writing of the text is very superficial.

4) 2.3 failed to reference the methodology used to carry out the work.

5) Tables 1 and 2 have blank "locations" (without values). If this happened in the research, I recommend using this symbol (-) in these locations, at least the readers will not get lost in understanding the data.

6) The conclusion is very superficial, it needs to be revised. It is important to close the work with a good conclusion. Please review this item.

7) The work to be published is of great importance to Toxics and the scientific community, but the way it is, with only 28 references, it is too few. This work should have at least 35 references.

8) Update the references, a lot of new research that should be cited.

After these corrections, I would like to see the work corrected and with a high chance of being accepted.

Author Response

Comments 1: Keywords need to start with a capital letter.

Response 1: Thanks for your suggestion. While in the template of the journal of Toxics, keywords were not asked to start with capital letters. We also have reviewed the latest pubulished articles in Toxics, and found that the keywords weren't started with capital letters.

Comments 2: The introduction has the most recent article in the year 2021. This cannot happen in the last 3 years (2022, 2023 and 2024) many researches were carried out with such proposed themes, please review this in detail, dear authors.

Response 2: Thanks for your opinion. In the last 3 years (2022, 2023 and 2024), there has not been significant progress in non-targeted analysis methods based on mass spectrometry (MS). Although there have been some researches reported on non-targeted screening of unknown disinfection by-products[1, 2], the method conducted in these researches were not selective for halogenated DBPs base on MS and couldn’t support the logic of introduction in this paper. Therefore, these progress in NTA method in the last 3 years were not included in the introduction. While we added several literatures published in the last 3 years in the DBPs development (line 34, line 38, and line 42 of page 1).

Comments 3: In item 2.2, I suggest an illustrative scheme of how the collection was done. The writing of the text is very superficial

Response 3: Thanks for your opinion. Our work aimed to develop a selective non-targeted method for exploring unknown halogenated DBPs, and the established method was applied in two tap waters and the corresponding source water. In item 2.2, we have described how the collection was done including sampling sites and sampling method. Due to limited space, the detail pretreatment procedures were demonstrated in Text S1 of supporting information.

Comments 4: 2.3 failed to reference the methodology used to carry out the work

Response4: Thank you for pointing this out, we have revised this part in line 109 and line 123-125 of page 3 with red font, and the reference was added in line 109.

Comments 5: Tables 1 and 2 have blank "locations" (without values). If this happened in the research, I recommend using this symbol (-) in these locations, at least the readers will not get lost in understanding the data

Response 5: Thanks for your good advice, we have filled the blank locations with symbol (-) in Table 1 and 2 with red font.

Comments 6: The conclusion is very superficial, it needs to be revised. It is important to close the work with a good conclusion. Please review this item

Response 6: Thank you for pointing this out. The conclusion has been revised in line 257-262 of page 9. As suggested by reviewer #1, a brief description of the method and the recommended operating parameters of the measuring device were added in the conclusion.

Comments 7: The work to be published is of great importance to Toxics and the scientific community, but the way it is, with only 28 references, it is too few. This work should have at least 35 references

Response 7: Thanks for your opinion. Due to limited space, we Due to the limited space, we put most of the work in structure speculation into supporting information which including 19 pages with 15 figures and 6 references. With the addition of several literatures from the past 3 years (2022, 2023 and 2024), the total number of references in the main text and supporting information has reached 39.

Comments 8: Update the references, a lot of new research that should be cited

Response 8: Thanks for your suggestion, a couple of new research has been cited and the references have been updated as mentioned in responses to comments 2 and 7.

Reviewer 3 Report

Comments and Suggestions for Authors

The manuscript entitled Selectively Non-Targeted Analysis of Halogenated Disinfection 2 Byproducts in Tap Water by Micro-LC QTOFMS represents valuable research that is not well exploited.

A similar LC-QTOF-MS method (applied for analyzing halogenated byproducts) was described by Gonzalez-Marino et al. (doi: 10.1016/j.watres.2011.10.027) since 2011. 

The manuscript in its present form is incomplete (cannot be discussed about chromatography, without a chromatogram in the manuscript), must be verified the reported data (i.e., the concentration of each haloacetic acids in standard material - the value of 1000 mg/mL cannot be real!), and contains some spelling mistakes (i.e., after ";" the authors must use lowercase!) - please verify the attached PDF.

I recommend that the authors improve this manuscript with some details about the performance parameters of the method and more samples (even the source water - surface water -, wastewater or bottled water).

Also, taking into account the journal, a small part of the manuscript can be dedicated to the toxicity of the halogenated DBPs or the health risks induced by the water consumption (if the samples contain halogenated DBPs).

My recommendation is to reject the manuscript (in this form) with the possibility of improvement and resubmission.

Author Response

Comments 1: The manuscript entitled Selectively Non-Targeted Analysis of Halogenated Disinfection 2 Byproducts in Tap Water by Micro-LC QTOFMS represents valuable research that is not well exploited.

Response 1: Thanks for your opinion. Our work aimed to develop a selective non-targeted method for exploring unknown halogenated DBPs, and the results testified that the established method was available to rapidly screen halogenated DBPs. The most difficult and time-consuming part was the resolving of MS data to speculate the structure of the screened DBPs. Due to the limited space, we put most of the work in structure speculation into supporting information which including 19 pages with 15 figures. From the attached file about review report provide by reviewer #3, we know that he/she didn’t find the supporting information in MDPI platform. Maybe it is the reason that he/she thought that this work was not well exploited.

Comments 2: A similar LC-QTOF-MS method (applied for analyzing halogenated byproducts) was described by Gonzalez-Marino et al. (doi: 10.1016/j.watres.2011.10.027) since 2011.

Response 2: I have downloaded this article written by Gonzalez-Marino et al [3]. This research presented an assessment of the sewage occurrence and biodegradability of seven parabens and three halogenated derivatives of methyl paraben (MeP). In this work, they used a LC-QTOF-MS to quantify the target compounds, and it was not similar to the functions of LC-QTOF-MS used in our work. In our work, we developed a selectively non-targeted analysis method for screening halogenated disinfection byproducts using multiple data acquisition modes of QTOF-MS.

Comments 3: The manuscript in its present form is incomplete (cannot be discussed about chromatography, without a chromatogram in the manuscript), must be verified the reported data (i.e., the concentration of each haloacetic acids in standard material - the value of 1000 mg/mL cannot be real!), and contains some spelling mistakes (i.e., after ";" the authors must use lowercase!) - please verify the attached PDF.

Response 3: Thanks for your kind suggestion. The chromatograph and chromatogram were presented in supporting information. The proper concentration unit of mg/L in haloacetic acids standard solution has been revised in line 80 of page 2. The other spelling mistakes listed in the attached PDF have been verified (line 52-56, line 82-88 of page 2; line 111, line 119, line 125 of page 3; line 158 of page 4).

Comments 4: I recommend that the authors improve this manuscript with some details about the performance parameters of the method and more samples (even the source water - surface water -, wastewater or bottled water).

Response 4: As the response to comments 1 above, our work aimed to develop a SNTA method for exploring unknown DBPs. The most difficult and time-consuming part was the resolving of MS data to speculate the structure of the screened DBPs. Due to the limited space, we put most of the work in structure speculation into supporting information which including 19 pages with 15 figures. We have provided instrumental parameters in manuscript and SPE parameters in supporting information. In future work, we will consider applying this method to various environmental water samples.

Comments 5: Also, taking into account the journal, a small part of the manuscript can be dedicated to the toxicity of the halogenated DBPs or the health risks induced by the water consumption (if the samples contain halogenated DBPs).

Response 5: The toxicity research needs standard chemicals of halogenated DBPs, however, many screened DBPs were possibly new compounds and couldn’t find commercial standard. There is another way to carry out the toxicity work by isolating corresponding DBPs using LC, while it is another big work needing more vigor and time. In consequent work, we would put in effort to carry out this work.

Round 2

Reviewer 2 Report

Comments and Suggestions for Authors

The article can be accepted.

Author Response

Thank you very much!

Reviewer 3 Report

Comments and Suggestions for Authors

I congratulate the authors on the corrections and supplementary material upload.

I think this version of the manuscript is better than the first one.

It can still be improved, but I know that the cost for such an analysis type is high.

As last recommendations:

1. the authors must verify the entire manuscript and verify the text color (i.e., lines 92-93, 131);

2. the reference lists (i.e., from the manuscript and supplementary material) do not respect the journal template (nor the text font).

Many thanks to the author for giving me a manuscript for a nice lecture! 

Author Response

Thanks for your suggestions, we would check carefully and revify the entire manuscritp including the references.